# Diagnostic accuracy of midkine on hepatocellular carcinoma: A meta-analysis

**Bo-han Zhang**, **Bo Li, Ling-xiang Kong, Lv-nan Yan, Jia-yin Yang** *

Department of Liver Surgery, Liver Transplantation Center, West China Hospital of Sichuan University, Chengdu, Sichuan, P. R. China

* docjackyang@163.com

**Data Availability Statement:** All relevant data are within the manuscript and its Supporting Information files.

**Funding:** This study was supported by grants from the 1.3.5 project for disciplines of excellence, West

## Abstract

### Objective

To evaluate the dependability and accuracy of midkine (MK) in the diagnosis of hepatocellular carcinoma (HCC).

### Methods

PubMed, EMBASE, Web of Science, China Biology Medicine disc and grey literature sources were searched from the date of database inception to January 2019. Two authors (B-H.Z. and B.L.) independently extracted the data and evaluated the study quality using the Quality Assessment of Diagnostic Accuracy Studies-2 tool. The sensitivity, specificity, positive likelihood ratio (LR+) and negative likelihood ratio (LR−) were estimated using a bivariate model. Moreover, hierarchical summary receiver operating characteristic curves were generated. The diagnostic odds ratio (DOR) and area under the curve (AUC) were pooled using a univariate model.

### Results

Nine articles (11 studies) were included (1941 participants). The bivariate analysis revealed that the sensitivity and specificity of MK for HCC diagnosis were 0.85 (95% CI 0.78–0.91) and 0.83 (95% CI 0.76–0.88), respectively. We also found a LR+ of 5.05 (95% CI 3.33–7.40), a LR− of 0.18 (95% CI 0.11–0.28), a DOR of 31.74 (95% CI 13.98–72.09) and an AUC of 0.91 (95% CI 0.84–0.99). Subgroup analyses showed that MK provided the best efficiency for HCC diagnosis when the cutoff value was greater than 0.5 ng/mL.

### Conclusions

MK has an excellent diagnostic value for hepatocellular carcinoma.

China Hospital, Sichuan University (ZY2017308) and the National Natural Science Foundation of China (No. 81470037 and 81770653). The funders had no role in study design, data collection and analysis, decision to publish, or preparation of the manuscript.

**Competing interests:** The authors have declared that no competing interests exist.

**Abbreviations:** AFP, α-fetoprotein; AUC, area under the curve; BGID, benign gastrointestinal disease; BLT, benign liver tumor; CBMdisc, China Biology Medicine disc; CHB, chronic hepatitis B; CHC, chronic hepatitis C; DOR, diagnostic odds ratio; DSL, DerSimonian and Laird; EIA, enzyme-linked immunoassay; ELISA, enzyme-linked immunosorbent assay; FN, false negative; FP, false positive; GIT, gastrointestinal tumor; HCC, hepatocellular carcinoma; HSROC, hierarchical summary receiver operating characteristic; $I^2$, inconsistency index; LR−, negative likelihood ratio; LR+, positive likelihood ratio; MK, midkine; PRISMA, Preferred Reporting Items for Systematic Reviews and Meta-Analyses; QUADAS, Quality Assessment of Diagnostic Accuracy Studies; TN, true negative; TP, true positive; US, ultrasound.

# Introduction

According to recent EASL HCC guidelines, approximately 854,000 new cases of liver cancer are diagnosed annually, among which hepatocellular carcinoma (HCC) is the most frequent type, accounting for up to 90 percent [1]. It is also the fifth most common cancer and the third most common cause of cancer-related death globally [2, 3]. The evolution of HCC is a multistep process from chronic liver disease to liver cirrhosis to primary HCC and eventually to metastatic HCC [4]. Patients who are diagnosed with HCC at an inchoate stage are more likely to be cured and have a 70% chance of living more than 5 years with the appropriate therapies such as hepatectomy or liver transplantation. Those who are diagnosed at an advanced stage, in contrast, qualify only for palliative treatments and have unsatisfactory median survival times ranging from 1 to 2 years [5]. These data corroborate the importance of early and accurate HCC diagnosis.

Some guidelines have ruled out α-fetoprotein (AFP) and recommend ultrasound (US) as the standard HCC monitoring procedure in cirrhotic patients [6, 7]. A recent meta-analysis concluded that US plus AFP may serve as an updated screening strategy for early HCC. However, the sensitivity and specificity are still low (63% and 45%, respectively) [5]. Moreover, many non-invasive screening tools, such as non-coding RNAs, des-γ-carboxyprothrombin and midkine (MK), have been investigated for use in the diagnosis of HCC [8]. As early as 1996, the serum level of MK assessed by enzyme-linked immunoassay (EIA) was found to be undetectable or lower than 0.6 ng/mL in healthy participants. However, more than fifty percent of HCC patients have an MK value varying from 0.6 to 8 ng/mL [9]. Using EIA, Ikematsu *et al* found that the highest level of normal serum MK does not reach 0.5 ng/mL, whereas the serum MK levels in 25 HCC cases were all greater than 0.5 ng/mL [10]. In addition, the secretory characteristic of MK makes it easy to quantitate in blood samples. All these characteristics indicate that MK has a promising future as a tool for non-invasive, early and sensitive HCC diagnosis [11]. However, the small number of cases in each study has limited the accuracy of the results, and the diagnostic ability of MK has not yet been fully elucidated. We conducted a systematic review and meta-analysis to determine the diagnostic power of MK for HCC.

# Methods

Drafted based on a preset protocol registered with PROSPERO 2018 (https://www.crd.york.ac.uk/PROSPERO/, CRD42018103537), the current meta-analysis was reported in accordance with the Preferred Reporting Items for Systematic Reviews and Meta-Analyses (PRISMA) statement (S1 Table) [12].

## Eligibility criteria

We enrolled studies that evaluated the use of the blood level of MK for the diagnosis of HCC. Studies with insufficient data or those including subjects with other types of liver tumours were excluded. If two studies had an identical cohort, we excluded the less informative one or the one with a smaller population.

## Identification and selection of studies

We systematically searched electronic databases including PubMed, EMBASE, Web of Science and China Biology Medicine disc (CBMdisc) from the data of database inception to January 2019, without imposing language restrictions. We used the MeSH terms "liver", "neoplasms", "carcinoma", "midkine", "sensitivity and specificity", "roc curve" and "diagnosis" for literature retrieval. Details of the search strategies for PubMed and EMBASE are presented in S1 Fig. For

CBMdisc, the combination of Chinese and English was required. Relevant unpublished work concerning MK and HCC was detected through a grey literature search of meeting proceedings and abstracts from the American Association for Cancer Research and American Society of Clinical Oncology. Finally, we identified candidate articles from the references of pertinent reviews and original studies.

First, the titles and abstracts of retrieved studies were independently screened and filtered by two investigators (B-H.Z. and B.L.). Second, the eligibility of the full-text articles was determined through separate scrutinization by two investigators. Duplicate use of an identical cohort was carefully evaluated. Disagreements were resolved through discussion or consultation with the third investigator (J-Y.Y.).

## Data extraction and quality assessment

Two investigators (B-H.Z. and B.L.) independently extracted information the below. First, the following main characteristics of the included studies were extracted: first author name, year of publication, country, sample type, number of participants, age, sex distribution, type of controls, detection method and cutoff values. Second, the following data concerning the diagnostic accuracy were collected: true positive (TP) rate, false positive (FP) rate, false negative (FN) rate, true negative (TN) rate, sensitivity and specificity. All data are publicly available in Open Science Framework (osf.io/gw8em/). The generic Quality Assessment of Diagnostic Accuracy Studies (QUADAS)-2 tool for diagnostic accuracy studies was applied for the quality evaluation of the enrolled studies [13]. Two investigators (B-H.Z. and B.L.) independently rated the four domains for the "Risk of Bias" and "Applicability Concerns". Consensus was reached through deliberation.

## Data synthesis

We fitted hierarchical models when there were at least 4 studies available. All calculations were accomplished with the package 'mada' in R (version 3.6.0). Cells in the contingency table that were zero needed a continuous correction with a recommended value of 0.5 for data analyses because certain ratios did not exist.

The sensitivity and specificity with corresponding 95% CIs were recalculated from the TP, FP, FN and TN rates extracted via a $2 \times 2$ table from each included study. The threshold effect was initially determined by the correlation between the sensitivity and false positive rate (1—specificity) through the visual evaluation of coupled forest plots and was further verified by the Spearman correlation coefficient $\rho$ ($> 0.6$) between the logit of sensitivity and the logit of the false positive rate [14].

The bivariate random effects model by Reitsma *et al.* [15] for diagnostic meta-analyses was applied to obtain the pooled estimates of the sensitivity, specificity, positive likelihood ratio (LR+) and negative likelihood ratio (LR−). Additionally, the hierarchical summary receiver operating characteristic (HSROC) curves were calculated with both the Rutter & Gatsonis and Rücker & Schumacher approaches [16, 17]. We implemented independent evaluations of the diagnostic performance based on the diagnostic odds ratio (DOR) using the DerSimonian and Laird (DSL) model [18] and the area under the curve (AUC) using Holling's model [19]. The heterogeneity of the DOR was determined using the chi-squared test and Higgins' inconsistency index ($I^2$). The statistic for the chi-squared test was Q, and a corresponding p value was calculated for the qualitative assessment of heterogeneity. We set 0.1 as the cutoff significance level [20]; however, with only 9 studies included in our investigation ($< 20$), the Q test should be interpreted very cautiously [21]. Higgins's $I^2$ statistic, calculated via the formula $I^2 = 100\% \times (Q—df)/Q$, was also calculated as a measure of between-study heterogeneity [22]. The level of

heterogeneity was deemed negligible, moderate, and considerable for $I^2$ values of 25%, 50%, and 75%, respectively [22]. We also conducted a series of prespecified subgroup analyses based on sample type, number of participants, country, control type and cutoff values. Two different thresholds (≤0.5 ng/mL and >0.5 ng/mL) were chosen for the exploration of diagnostic accuracy in reference to the existing practice [10]. Deeks' funnel plot was generated to test for publication bias [23].

## Results

### Study selection and characteristics

As seen in the flowchart, a total of 139 articles met the preliminary standards, including 41 from PubMed, 42 from EMBASE, 28 from Web of Science and 28 from CBMdisc (Fig 1). Ninety-three records remained after removing duplicates. Additionally, 55 irrelevant studies and 10 reviews and meta-analyses were excluded based on screening the titles and abstracts. The remaining 28 articles were considered eligible for full-text review. Nineteen additional studies comprising 16 with insufficient data, 2 with identical cohorts and 1 with a case group composed of patients with cholangiocarcinoma were excluded. A manual search for grey literature and references found no applicable results. These strict eliminations yielded a group of 9 articles (11 studies) for inclusion in the meta-analysis [24–32], one of which was a poster presentation [24]. The studies were conducted in China, Egypt, Taiwan and Australia.

The primary attributes of the enrolled studied were summarized and are listed alphabetically in Table 1 and S2 Table. Six studies also analysed the diagnostic potential of AFP [24, 25, 28, 30–32], and only three studies addressed the combined diagnostic potential of AFP and MK [28–30]. Due to the scarcity of studies, we did not calculate the indexes relating to the combined AFP and MK group. The number of participants in each study ranged from 70–833, with a median of 164. In total, the meta-analysis included 1941 individuals, namely, 834 HCC patients and 1107 non-HCC participants. Specifically, the non-HCC participants included 123 with gastrointestinal tumour (GIT), 73 with benign liver tumour (BLT), 453 with liver cirrhosis, 27 with chronic hepatitis C (CHC), 86 with chronic hepatitis B (CHB), 50 with benign gastrointestinal disease (BGID) and 295 healthy people. Enzyme-linked immunosorbent assay (ELISA) served as the uniform testing method [24–28, 30–32] for serum MK. Only one study investigated the MK level in whole blood, and they performed the experiment with TaqMan [29].

### Quality assessment

The results of the QUADAS-2 assessment regarding the risk of bias and applicability concerns are summarized in S3 Table. We did not assign quality scores because of underlying heterogeneity [33].

The details are presented below: for the "risk of bias", the major concerns were "patient selection" and the "index test". This was mainly due to the uncertainty of whether consecutive or random sample collection was used, the case-control design, the arbitrary use and absence of a preset cutoff value. In the absence of explicit reference standards, two studies were marked as high risk. In addition, studies without the presentation of an appropriate interval between the index test and the reference standard were deemed unclear or risky. With regard to the "applicability concerns", most of the included studies showed low risk, and the two unclear risk studies did not describe the reference standard; hence, we could not evaluate the applicability.

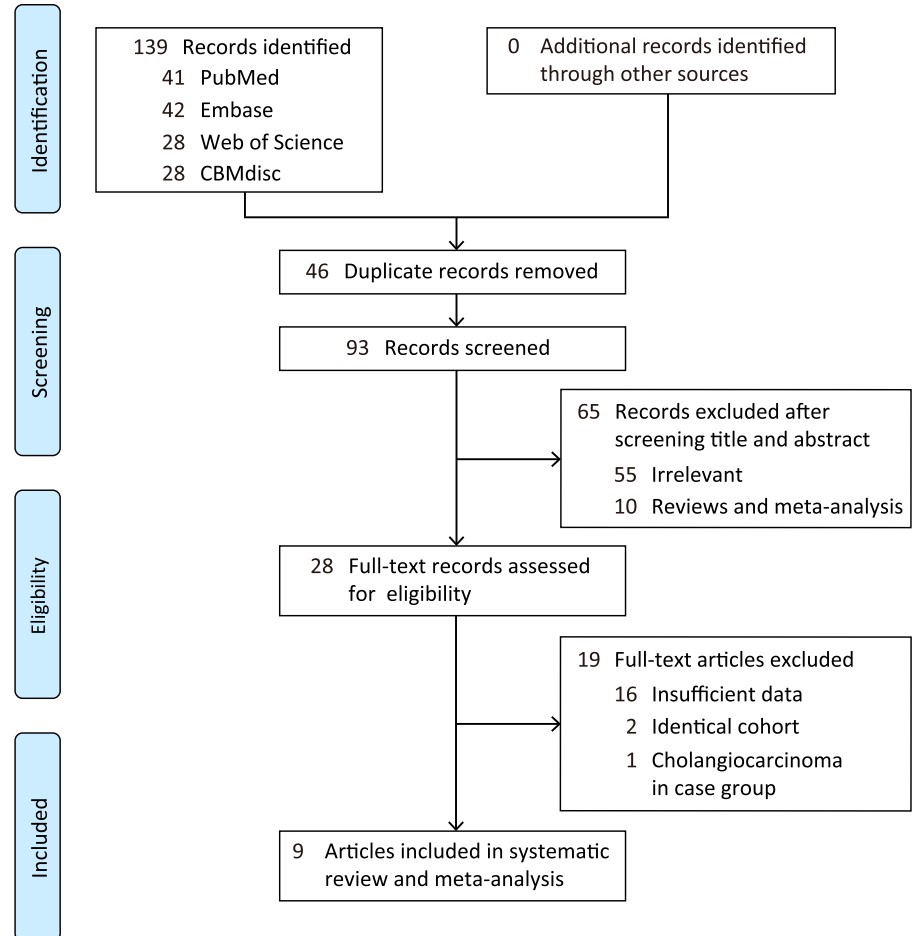

**Fig 1. Flow diagram of the literature search process and study inclusion.**

## Diagnostic accuracy

In general, our analysis revealed that the sensitivity and specificity of MK in the diagnosis of HCC ranged from 0.60 to 1.00 (median, 0.87) and from 0.62 to 1.00 (median, 0.84), respectively (Fig 2A). Neither the visual assessment of the coupled forest plots nor the Spearman correlation coefficient ρ (-0.50, 95% CI -0.85–0.14) supported the threshold effect. For AFP, we incorporated common cutoff values (20, 40 and 200 ng/mL) among the various values addressed by one study for further analysis. The sensitivity and specificity ranged from 0.25 to 0.83 (median, 0.52) and from 0.35 to 1.00 (median, 0.84), respectively (Fig 2B). No threshold effect was found on the forest plots or with the Spearman ρ (0.38, 95% CI -0.45–0.85).

For MK, the pooled sensitivity and specificity were 0.85 (95% CI 0.78–0.91) and 0.83 (95% CI 0.76–0.88), respectively. The sensitivity was statistically superior to that of AFP (p = 0.000), which was only 0.53 (95% CI 0.43–0.64). However, AFP had a slightly better specificity (0.84, 95% CI 0.64–0.94) compared with MK, although the difference was nonsignificant (p = 0.818). We also found a LR+ of 5.05 (95% CI 3.33–7.40) and a LR− of 0.18 (95% CI 0.11–0.28) for MK, and a LR+ of 3.79 (95% CI 1.62–8.25) and a LR− of 0.56 (95% CI 0.46–0.69) for AFP. The Rutter & Gatsonis and Rücker & Schumacher HSROC curves for MK and AFP are shown in Fig 3. Scattered circles represent individual studies; summary estimates originating from the

**Table 1. Characteristics of included studies.**

| Author | Year | Country | Sample | Case (HCC) | | | Control | | | | Method | Cut-off (ng/mL) |
|---|---|---|---|---|---|---|---|---|---|---|---|---|
| | | | | No. | Age* | F/M | No. | Age* | F/M | Type | | |
| Habachi *et al* | 2018 | Egypt | Serum | 86 | - | - | 89 | - | - | 89 Cirrhosis | ELISA | 5.1 |
| Hodeib *et al* | 2017 | Egypt | Serum | 35 | 49.10±4.60 | 9/26 | 35 | 48.00 ±4.40 | 11/24 | 35 Normal | ELISA | 0.65 |
| Hung *et al* | 2011 | Taiwan | Serum | 72 | - | - | 120 | - | - | 54 GIT, 6 BLT, 10 Cirrhosis, 50 BGID | ELISA | 0.5 |
| Li *et al* | 2006 | China | Serum | 104 | - | - | 60 | - | - | 20 BLT, 20 Cirrhosis, 20 Normal | ELISA | 0.07 |
| Mashaly *et al* | 2018 | Egypt | Serum | 44 | 58.11±1.05 | 11/33 | 31 | 56.55 ±1.37 | 13/18 | 31 Cirrhosis | ELISA | 1.683 |
| Saad *et al* | 2013 | Egypt | Blood | 29 | 55.60±7.90 | 9/20 | 45 | - | 14/31 | 18 Cirrhosis, 27 CHC | TaqMan | - |
| Shaheen *et al* | 2015 | Egypt | Serum | 40 | 52 | 10/30 | 30 | 48 | 12/18 | 30 Cirrhosis | ELISA | 0.387 |
| | | | Serum | 40 | 52 | 10/30 | 30 | 45 | 13/17 | 30 Normal | ELISA | - |
| Vongsuvanh *et al* | 2016 | Australia | Serum | 86 | 62.20±11.40 | 11/75 | 172 | - | 22/ 150 | 86 Cirrhosis, 86 CHB | ELISA | 0.44 |
| Zhu *et al* | 2013 | China | Serum | 252 | <50(99), ≥50 (153) | 33/ 219 | 455 | - | - | 69 GIT, 47 BLT, 129 Cirrhosis, 210 Normal | ELISA | 0.654 |
| | | | Serum | 86 | <50(35), ≥50 (51) | 18/68 | 40 | - | - | 40 Cirrhosis | ELISA | 0.654 |

Abbreviations: HCC hepatocellular carcinoma, F/M female versus male, ELISA enzyme-linked immunosorbent assay, GIT gastrointestinal tumor, BLT benign liver tumor, BGID benign gastrointestinal disease, CHC chronic hepatitis C, CHB chronic hepatitis B, SD standard deviation.

*Numbers were presented as mean±SD, median or range.

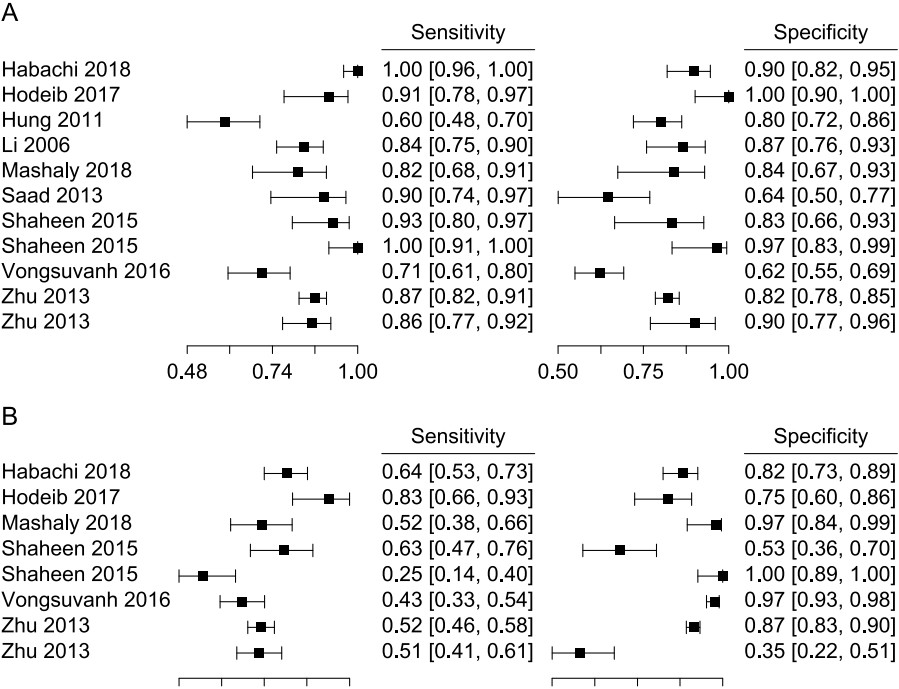

**Fig 2. Forest plots of (A) MK and (B) AFP.** High degree of heterogeneities for both sensitivity and specificity estimates were obtained.

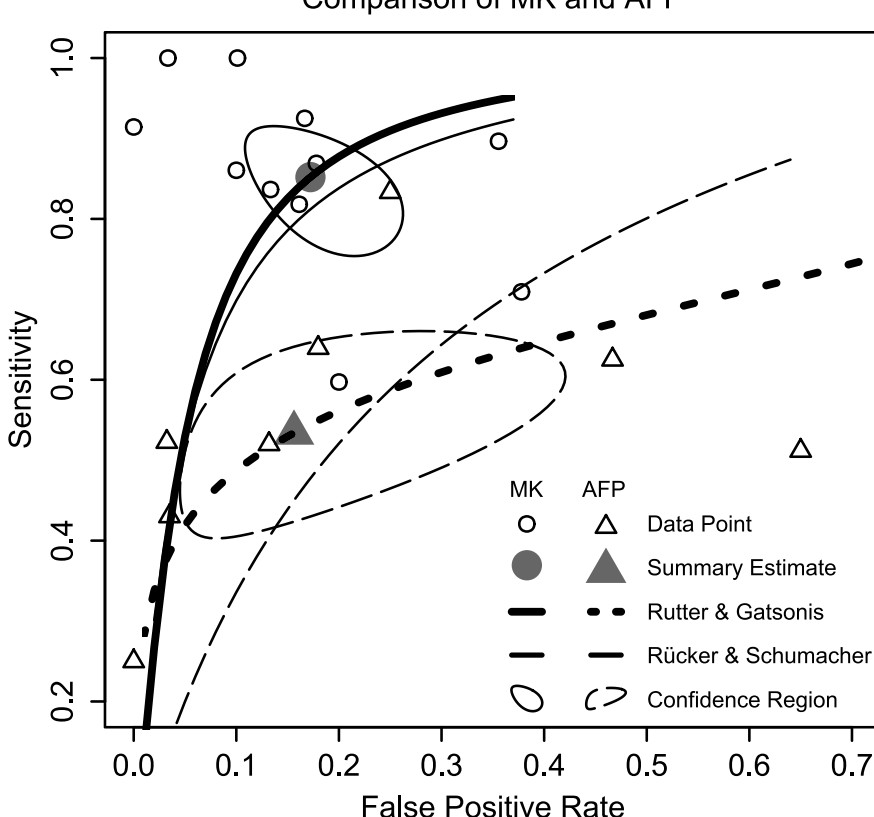

**Fig 3. Diagnostic accuracy comparison between MK and AFP using HSROC curves.**

bivariate model as well as the 95% confidence intervals were also plotted. The data points for MK are clustered in the top left corner, while the data points for AFP are mostly located in the middle left of the plot, indicating that MK is more sensitive than AFP for the diagnosis of HCC. Meanwhile, their similar projection positions on the X-axis indicate that the specificities are comparable. The pooled DORs were 31.74 (95% CI 13.98–72.09) for MK and 6.21 (95% CI 2.62–14.69) for AFP. MK studies had moderate heterogeneity of DOR (Q = 16.00, p = 0.10, $I^2$ = 37.52%), and AFP studies had negligible heterogeneity of DOR (Q = 7.85, p = 0.35, $I^2$ = 10.83%). The pooled AUCs were 0.91 (95% CI 0.84–0.99) for MK and 0.71 (95% CI 0.53–1.00) for AFP. Evidently, MK had better discriminatory power than AFP to distinguish HCC from non-HCC. The results of the subgroup analyses for MK studies are shown in Table 2. A symmetric funnel plot (Fig 4) showed no publication bias (p = 0.37) in the included studies according to the method of Deeks *et al.* [23].

## Discussion

The abnormal expression of MK has been widely investigated in various malignancies [32, 34]. In contrast, MK is rarely detectable in non-malignant blood samples, and the encouraging non-invasive diagnostic potential of MK for tumours is worth in-depth investigation. Jing *et al* concluded that MK has great performance in the diagnosis of malignant diseases such as oesophageal squamous cell carcinoma, paediatric embryonal tumour, colorectal cancer, hepatocellular carcinoma, thyroid cancer, non-small cell lung cancer, mesothelioma and head and neck squamous cell carcinoma. However, tumour heterogeneity confers substantial limitations on

**Table 2. Synopsis of results from subgroup analyses depending on sample type, number of participants, country, control type and cutoff values used for the diagnosis of hepatocellular carcinoma.**

| Characteristics | Studies, No. | Participants, No. | Sensitivity, (95% CI) | Specificity, (95% CI) | Positive Likelihood Ratio (95% CI) | Negative Likelihood Ratio (95% CI) | Diagnostic Odds Ratio (95% CI) | Area Under the Curve (95% CI) |
|---|---|---|---|---|---|---|---|---|
| Sample serum | 10 | 1907 | 0.85 (0.77–0.91) | 0.84 (0.77–0.89) | 5.52 (3.55–8.23) | 0.18 (0.10–0.29) | 35.64 (14.63–86.83) | 0.91 (0.83–1.00) |
| Sample blood | 1 | 74 | NA[a] | NA[a] | NA[a] | NA[a] | NA[a] | NA[a] |
| Participants ≤100 | 5 | 359 | 0.88 (0.82–0.93) | 0.86 (0.69–0.94) | 6.96 (2.72–15.80) | 0.14 (0.08–0.24) | 58.27 (15.03–225.90) | 0.96 (0.93–1.00) |
| Participants >100 | 6 | 1622 | 0.82 (0.73–0.90) | 0.82 (0.70–0.88) | 4.63 (2.69–7.43) | 0.23 (0.12–0.40) | 22.20 (7.77–63.40) | 0.88 (0.78–1.00) |
| Country Egypt | 6 | 534 | 0.91 (0.83–0.96) | 0.85 (0.73–0.92) | 6.56 (3.29–12.20) | 0.11 (0.05–0.21) | 97.73 (22.69–421.04) | 0.98 (0.97–1.00) |
| Country China | 3 | 997 | NA[a] | NA[a] | NA[a] | NA[a] | NA[a] | NA[a] |
| Country others | 2 | 450 | NA[a] | NA[a] | NA[a] | NA[a] | NA[a] | NA[a] |
| Control CLD[b] | 6 | 778 | 0.85 (0.76–0.90) | 0.79 (0.68–0.87) | 4.17 (2.44–6.80) | 0.20 (0.11–0.34) | 28.79 (7.92–104.65) | 0.91 (0.82–1.00) |
| Control normal | 2 | 140 | NA[a] | NA[a] | NA[a] | NA[a] | NA[a] | NA[a] |
| Control mixed | 3 | 1063 | NA[a] | NA[a] | NA[a] | NA[a] | NA[a] | NA[a] |
| Cut off ≤0.5ng/mL | 4 | 684 | 0.78 (0.61–0.89) | 0.79 (0.66–0.88) | 3.81 (1.96–6.77) | 0.30 (0.14–0.55) | 12.93 (4.21–39.71) | 0.84 (0.75–0.95) |
| Cut off >0.5ng/mL | 5 | 1153 | 0.87 (0.83–0.90) | 0.86 (0.80–0.90) | 6.19 (4.31–8.74) | 0.16 (0.12–0.20) | 57.08 (21.09–154.48) | 0.95 (0.92–0.99) |
| Cut off none | 2 | 144 | NA[a] | NA[a] | NA[a] | NA[a] | NA[a] | NA[a] |

Abbreviations: CLD, chronic liver disease; NA, not available.

[a]Insufficient data for pooling results.

[b]Including cirrhosis, chronic hepatitis B and chronic hepatitis C.

the conclusions [35]. Here, we found a "good" AUC for MK, compared with a "reasonable" AUC for AFP according to the criterion proposed by Jones *et al.* [36]. Likewise, the pooled DOR for MK eclipsed the one for AFP. The overall sensitivity was greater for MK than for AFP ($p = 0.000$), yet the overall specificity was approximately equal. In summary, MK is an adequate diagnostic biomarker that is generally more sensitive than AFP for the discrimination of HCC patients from normal individual and cirrhosis, CHC, CHB, GIT, BLT and BGID patients.

To the best of our knowledge, this is the first systematic review and meta-analysis evaluating the diagnostic accuracy of MK in HCC individuals. We conducted the current systematic review according to the PRISMA guidelines [12] and used a preestablished protocol registered in PROSPERO to guarantee the internal validity of our conclusions. A rigorous search of online databases and grey literature sources without language restriction avoided selection bias stemming from the source of the literature. Two authors (B-H.Z. and B.L.) independently extracted data and assessed the quality of the studies using QUADAS-2 [13], a meticulous tool for diagnostic meta-analyses. We used both univariate and bivariate models to synthesize the existing data.

Nine articles including 11 studies were collected and included in the subgroup analyses of MK. We incorporated five covariates: sample type, number of participants, country, control type and cutoff value. As indicated, the pooled sensitivity of the studies with >100 participants [24, 26, 27, 31, 32] was lower than that of studies with ≤100 participants [25, 28–30]. In addition, the pooled specificity of studies with >100 participants was lower than that of studies with ≤100 participants. We noticed that the entire population of studies with ≤100

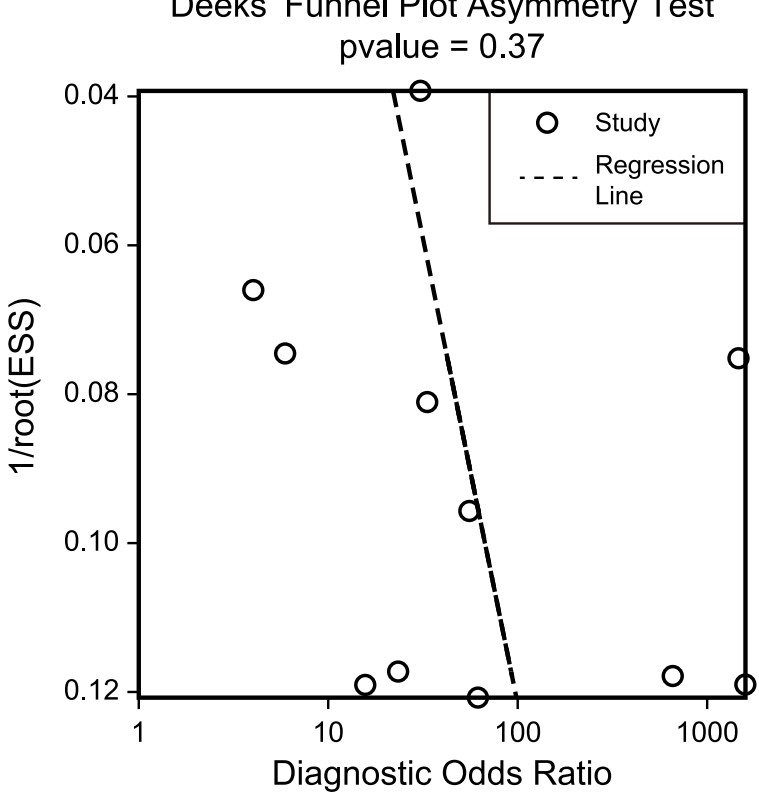

**Fig 4. Funnel plot over included studies according to Deeks *et al*.**

participants was still small, with the maximum sample size of only 75 [28]. However, as we know, the small-study effect is a typical mechanism well documented in randomized clinical trial studies, and it seems less marked in diagnostic meta-analyses [37]. Furthermore, the pooled sensitivity and specificity of MK in studies with cutoff values >0.5 ng/mL were manifestly greater than those in studies with cutoff values ≤0.5 ng/mL (0.87 *versus* 0.78 and 0.86 *versus* 0.79, respectively). Similarly, we found greater LR+ (6.19 *versus* 3.81), DOR (57.08 *versus* 12.93) and AUC (0.95 *versus* 0.84) values and lower LR− (0.16 *versus* 0.30) values. These results corroborate that the use of cutoff values >0.5 ng/mL resulted in the best diagnostic performance. Li *et al* used heparin-ELISA to determine MK expression [27]. We noticed that MK can bind to heparin sulfate on the vascular endothelial surface. This combination could undermine the sensitivity and specificity of the use of serum MK for HCC detection. As reported previously, the intravenous administration of heparin could increase the serum MK level in a dose-dependent manner [26]. A heparin-ELISA is another type of heparin test that increases the sensitivity of MK, or in other words, lowers the cutoff value (0.07 ng/mL).

Traditional HSROC parametrization (Rutter & Gatsonis method) revealed the conspicuous superiority of MK over AFP with regard to the diagnosis of HCC. It should also be noted that in this meta-analysis, an enrolled study represented the particular population of a single institution and consequently defined flexible optimal cutoff values. The diagnostic efficiency per study could be overestimated, correspondingly increasing the power of the pooled estimates to a certain degree. In this case, an alternative approach, the conservative Rücker & Schumacher method, was employed to compute the HSROC curve, acting as a supplement to account for this tiny flaw. The resulting curves all verified the better diagnostic accuracy of MK compared

with AFP. The rate of AFP-negative (<20 ng/mL) HCC limits the practicability of AFP for HCC surveillance. The secretory ability of hepatic tumours could be dampened by their small size. Even among larger lesions, twenty percent are not correlated with upregulated levels of AFP [31]. Five studies agree that the MK level is independent of AFP level [28–30, 32, 38]. Additionally, four studies reported a high positivity rate for MK in AFP-negative HCC [24, 28, 31, 32], suggesting the excellent sensitivity of the combination of MK and AFP. In addition, Vongsuvanh et al suggest the capacity of MK to be used for the pre-clinical diagnosis of HCC. In 2000, Ikematsu and colleagues addressed the decreased level of serum MK in 4 out of 5 HCC patients after curative surgery [10]. A later study reported that thirty-six HCC patients had experienced a sharp decline in the serum level of MK four weeks after hepatectomy. Meanwhile, the serum levels of MK in patients with documented recurrence (20/36) increased to the preoperative levels [32]. However, Hung et al concluded that the longitudinal monitoring of serum MK is incapable of detecting HCC recurrence and de novo HCC [26]. Further well-designed studies with larger sample sizes are needed to settle those disputes.

Limitations should be acknowledged. First, with an exhaustive search procedure, only 9 eligible articles (11 studies) were obtained. Quality assessment uncovered studies with high or unclear risks of bias. This could be explained by their suboptimal study designs. Second, only three studies reported or had sufficient information to calculate the data regarding the diagnostic accuracy of combined MK and AFP; hence, we could not perform a comparative study of the combined and individual diagnostic accuracies. The lack of AFP studies in the included literature and the selection of different cut-off values for AFP may also undermine the stability of our results. Third, the diversity of the control group weakened the accuracy of the specificity values. Specifically, direct-acting antiviral agents (DAA) and nucleotide analogues (NUC) are safe and effective at eradicating HCV and HBV infection, respectively. Therefore, the possible use of DAA or NUC regimens in patients with CHC and CHB in the control group may impede a robust conclusion. Likewise, the aetiology of liver cirrhosis and the trend for the application of lower AFP thresholds (<20 ng/mL) to monitor HCC recurrence may affect the robustness of the conclusion.

In conclusion, MK has a high diagnostic accuracy for HCC screening. More studies are needed to investigate the differential expression of MK in blood samples from patients with different degrees of liver fibrosis and its value in the diagnosis of cirrhotic and non-cirrhotic liver cancer patients. Whether the combination of MK and AFP provides better performance for HCC detection remains unknown. Further studies with rigorous designs are warranted to complete a full-scale evaluation of combined MK and AFP implementation as a means to accelerate the clinical investigation of individualized screening options.

## Supporting information

**S1 Fig. Search strategy.**
(DOCX)

**S1 Table. PRISMA 2009 checklist.**
(DOC)

**S2 Table. Diagnostic accuracy of the included studies.**
(DOCX)

**S3 Table. Quality assessment for 9 studies using QUADAS-2.**
(DOCX)

## Acknowledgments

We thank all authors who provided published information for our meta-analysis.

## Author Contributions

**Conceptualization:** Bo-han Zhang, Lv-nan Yan, Jia-yin Yang.

**Data curation:** Bo-han Zhang, Bo Li, Ling-xiang Kong, Jia-yin Yang.

**Formal analysis:** Bo-han Zhang, Bo Li.

**Funding acquisition:** Jia-yin Yang.

**Investigation:** Bo-han Zhang, Bo Li.

**Methodology:** Bo-han Zhang, Jia-yin Yang.

**Software:** Bo-han Zhang, Bo Li, Ling-xiang Kong.

**Supervision:** Lv-nan Yan, Jia-yin Yang.

**Validation:** Ling-xiang Kong.

**Visualization:** Bo-han Zhang.

**Writing – original draft:** Bo-han Zhang.

**Writing – review & editing:** Bo-han Zhang, Jia-yin Yang.

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
