## [Decision Letter · Decision Letter 0]

8 Jul 2019

PONE-D-19-15810

Diagnostic accuracy of midkine on hepatocellular carcinoma: a meta-analysis

PLOS ONE

Dear Dr. Jia-yin Yang,

Thank you for submitting your manuscript to PLOS ONE. After careful consideration, we feel that it has merit but does not fully meet PLOS ONE’s publication criteria as it currently stands. Therefore, we invite you to submit a revised version of the manuscript that addresses the points raised during the review process.

ACADEMIC EDITOR: Although it is of interest, the reviewers have raised a number of points which we believe major modifications are necessary to improve the manuscript, taking into account the reviewers' remarks. 

We would appreciate receiving your revised manuscript by Aug 22 2019 11:59PM. To enhance the reproducibility of your results, we recommend that if applicable you deposit your laboratory protocols in protocols.io, where a protocol can be assigned its own identifier (DOI) such that it can be cited independently in the future. For instructions see: http://journals.plos.org/plosone/s/submission-guidelines#loc-laboratory-protocols

We look forward to receiving your revised manuscript.

Kind regards,

Wisit Cheungpasitporn, MD, FACP

University of Mississippi Medical Center

Twitter: @wisit661 Email: wcheungpasitporn@gmail.com 

Academic Editor

PLOS ONE

Journal Requirements:

Reviewers' comments:

Reviewer's Responses to Questions

**Comments to the Author**

1. Is the manuscript technically sound, and do the data support the conclusions?

Reviewer #1: Partly

Reviewer #2: Yes

Reviewer #3: Yes

2. Has the statistical analysis been performed appropriately and rigorously? 

Reviewer #1: Yes

Reviewer #2: Yes

Reviewer #3: Yes

3. Have the authors made all data underlying the findings in their manuscript fully available?

Reviewer #1: Yes

Reviewer #2: Yes

Reviewer #3: Yes

4. Is the manuscript presented in an intelligible fashion and written in standard English?

Reviewer #1: Yes

Reviewer #2: Yes

Reviewer #3: No

5. Review Comments to the Author

Reviewer #1: Dr. Zhang and colleagues made a comprehensive review of literature on accuracy of midkine for the diagnosis of HCC.

However,

- Remove “holistic” from introduction section

- In the result section, authors decribe the control group without HCC as composed by 113 patients with chronic hepatitis C or HBV, without comments on possible DAA or NUC treatment, and by 453 subjects with liver cirrhosis, without specify the etiology of liver disease. In discussion section, this limit should be pointed out and highlighted, as well as the AFP levels used (20-200 ng/mL) for the accuracy comparison are very hight. According to literature, accuracy of AFP for HCC changes according to etiology, and nowadays lower AFP thresholds are under evaluation for increasing HCC surveillance.

- Add recent EASL HCC guidelines in references section and introduction

- Only 6 studies evaluated the role of AFP vs MK, and only three AFP+MK: this point of limit numbers of studies on AFP, and moreover with different cut-off on etiology should be stressed in discussion, and also in results section of the abstract. In fact, here there are no data on AFP and MK comparison in result section, but AFP appears in the conclusion statement. Modify results section or change conclusion statement, removing AFP.

Reviewer #2: This study investigated about the dependability and accuracy of midkine (MK) to diagnose HCC by systematic review and meta- analysis. In this study, 9 articles were incorporated and the authors reported that the sensitivity and specificity of MK for diagnosing HCC were 0.85 and 0.93, respectively. And furthermore, the authors indicate that the best efficient cut-off value of MK level to diagnose HCC was 0.5ng/mL.

Although not entirely novel, this study is potentially interesting to the readers as the authors showed the possibility of MK as a novel diagnostic biomarker of HCC. The study design is well organized and the statistics seem to have been conducted appropriately.

However, I have several points as indicated below need to be addressed in the manuscript by the authors to improve the quality of this article.

Major points

1. According to the past report by Hung et al, serum MK levels could have potential to be useful to monitor HCC progression. Have you analyzed about the correlation between the HCC stage or prognosis of the patients with HCC and serum MK levels? Of course, I know that this research focused only about the diagnostic usefulness of MK. But if the authors have some informative data about the possibility of MK for prediction of prognosis or monitoring the response to therapy, it would be more useful to include the information about this point.

2. Is there any difference in MK level according to the liver fibrosis status? Is the cut-off level to diagnose of HCC the same between the cirrhosis and non-cirrhosis patients? Whether the authors have some data or not, it would be useful to mention about this point in result or discussion part.

Minor point

1. “heparin” in the last line of page 20 is misspelled.

Reviewer #3: Please also include timeline of the literature search in the method section of the abstract.

Please also include timeline of the literature search in the method section of the Fulltext.

When Pubmed is used for the search, MESH terms are always recommended to be included.

Search terms in PubMed and Embase are different. Please attach syntax used in each database as supplementary.

It will be better to show kappa for the selection and data extraction. Please show the data of kappa of agreement during the systematic searches. How disagreements were solved during the systematic search among two independent reviewers?

Figure1, suggest to use PRISMA 2009 Flow Diagram platform

Please make the data for this review publicly available, possibly through the Open Science Framework (osf.io). Items to include: list of excluded studies, commands for statistical analysis, spreadsheets or data used for the meta-analyses, etc. Making data publicly available will promote the reproducibility of the review and is best practices for systematic reviews and meta-analyses.

Some revision of the English language is needed. There are some parts of the paper where it is quite difficult to make sense of some sentences. English edit will help to improve the quality of the manuscript.

“makes it readily to quantitate in blood samples” is not correct in grammar.

“above fifty percent HCC patients” is not correct

“Studies that with insufficient data” is not correct

“other type” is not correct in grammar.

“10 review and meta-analysis were excluded based upon” is not correct in grammar.

“literatures” is not correct in grammar. It is uncountable noun

“Jing et al concludes that MK demonstrates great performance in the diagnosis of malignant diseases like esophageal squamous cell carcinoma” is not correct. It should be past tense.

“Hung et al concludes that the longitudinal monitoring of serum MK is incapable of detecting HCC recurrence and de novo HCC” is not correct. It should be past tense.

“A recent meta-analysis concludes that US plus AFP may serve as an updated screening strategy

for early HCC.” is not correct. It should be past tense.

It is not professional to use “And” at the beginning of sentences in academic writing.

6. PLOS authors have the option to publish the peer review history of their article (what does this mean?). If published, this will include your full peer review and any attached files.

Reviewer #1: No

Reviewer #2: No

Reviewer #3: No

---

## [Author Response · Author response to Decision Letter 0]

20 Aug 2019

To editor, 

I sincerely appreciate your hard work.

---

## [Decision Letter · Decision Letter 1]

2 Sep 2019

[EXSCINDED]

PONE-D-19-15810R1

Diagnostic accuracy of midkine on hepatocellular carcinoma: a meta-analysis

PLOS ONE

Dear Jia-yin Yang,

Thank you for submitting your manuscript to PLOS ONE. After careful consideration, we feel that it has merit but does not fully meet PLOS ONE’s publication criteria as it currently stands. Therefore, we invite you to submit a revised version of the manuscript that addresses the points raised during the review process.

ACADEMIC EDITOR: The reviewer (s) have still raised a number of points which we believe major modifications are necessary to improve the revised manuscript, taking into account the reviewers' remarks.  Please consider and address each of the comments raised by the reviewers before resubmitting the manuscript. This letter should not be construed as implying acceptance, as a revised version will be subject to re-review.

We would appreciate receiving your revised manuscript by Oct 17 2019 11:59PM. To enhance the reproducibility of your results, we recommend that if applicable you deposit your laboratory protocols in protocols.io, where a protocol can be assigned its own identifier (DOI) such that it can be cited independently in the future. For instructions see: http://journals.plos.org/plosone/s/submission-guidelines#loc-laboratory-protocols

We look forward to receiving your revised manuscript.

Kind regards,

Wisit Cheungpasitporn, MD, FACP

Academic Editor

PLOS ONE

Reviewers' comments:

Reviewer's Responses to Questions

**Comments to the Author**

1. If the authors have adequately addressed your comments raised in a previous round of review and you feel that this manuscript is now acceptable for publication, you may indicate that here to bypass the “Comments to the Author” section, enter your conflict of interest statement in the “Confidential to Editor” section, and submit your "Accept" recommendation.

Reviewer #1: All comments have been addressed

Reviewer #2: All comments have been addressed

Reviewer #3: (No Response)

2. Is the manuscript technically sound, and do the data support the conclusions?

Reviewer #1: Yes

Reviewer #2: Yes

Reviewer #3: Partly

3. Has the statistical analysis been performed appropriately and rigorously? 

Reviewer #1: Yes

Reviewer #2: Yes

Reviewer #3: Yes

4. Have the authors made all data underlying the findings in their manuscript fully available?

Reviewer #1: Yes

Reviewer #2: Yes

Reviewer #3: No

5. Is the manuscript presented in an intelligible fashion and written in standard English?

Reviewer #1: Yes

Reviewer #2: Yes

Reviewer #3: No

6. Review Comments to the Author

Reviewer #1: (No Response)

Reviewer #2: Some comments this reviewer pinted out were properly addressed. This revewer think the munuscript should be accepted for publication.

Reviewer #3: The authors claimed they included "MESH terms in our searchsyntax." However, this needs to be mentioned formally in the method section of manuscript. MESH term for Pubmed should be provided in the full manuscript.

The authors claimed "We attached the syntax used in PubMed and Embase as supplementary." However, this is not good enough. Attached file S1 Figure.esp is not professional. This needs to be more formal in the manuscript for as .doc that easy to assess.

The authors claimed they made "data publicly available through Open Science Framework"; however, they have not provided the formal like that can access data publicly in the manuscript. Need to include in method section or the first part of result.

The authors claimed they made changes in English edits as "These mistakes were carefully corrected." However, there arevery minor changes in the manuscript and still very difficult to read through in English writing. English language Edit is needed including formal proof.

If the investigators do not take these comments seriously to improve manuscript, I suggest rejection for this submission.

7. PLOS authors have the option to publish the peer review history of their article (what does this mean?). If published, this will include your full peer review and any attached files.

Reviewer #1: No

Reviewer #2: No

Reviewer #3: No

---

## [Author Response · Author response to Decision Letter 1]

12 Sep 2019

We thank the reviewers for their valuable suggestions.

---

## [Decision Letter · Decision Letter 2]

24 Sep 2019

Diagnostic accuracy of midkine on hepatocellular carcinoma: a meta-analysis

PONE-D-19-15810R2

Dear Dr. Jia-yin Yang,

We are pleased to inform you that your manuscript has been judged scientifically suitable for publication and will be formally accepted for publication once it complies with all outstanding technical requirements.

With kind regards,

Wisit Cheungpasitporn, MD, FACP

University of Mississippi Medical Center

Twitter: @wisit661 Email: wcheungpasitporn@gmail.com 

Academic Editor

PLOS ONE

Additional Editor Comments:

I want to commend the authors on their superb efforts to revise the manuscript according to all reviewers’ suggestions. The quality of the manuscript has improved substantially.

Reviewers' comments:

Reviewer's Responses to Questions

**Comments to the Author**

1. If the authors have adequately addressed your comments raised in a previous round of review and you feel that this manuscript is now acceptable for publication, you may indicate that here to bypass the “Comments to the Author” section, enter your conflict of interest statement in the “Confidential to Editor” section, and submit your "Accept" recommendation.

Reviewer #1: All comments have been addressed

Reviewer #2: All comments have been addressed

Reviewer #3: All comments have been addressed

2. Is the manuscript technically sound, and do the data support the conclusions?

Reviewer #1: Yes

Reviewer #2: Yes

Reviewer #3: Yes

3. Has the statistical analysis been performed appropriately and rigorously? 

Reviewer #1: Yes

Reviewer #2: Yes

Reviewer #3: Yes

4. Have the authors made all data underlying the findings in their manuscript fully available?

Reviewer #1: Yes

Reviewer #2: Yes

Reviewer #3: Yes

5. Is the manuscript presented in an intelligible fashion and written in standard English?

Reviewer #1: Yes

Reviewer #2: Yes

Reviewer #3: Yes

6. Review Comments to the Author

Reviewer #1: (No Response)

Reviewer #2: Authors have fully modified their manuscript. I think this manuscript should be accepted for publication.

Reviewer #3: All my concerns have been fully elucidated, missing sections and analyses have been completed. Finally, comprehension errors have been corrected. Good work!

7. PLOS authors have the option to publish the peer review history of their article (what does this mean?). If published, this will include your full peer review and any attached files.

Reviewer #1: No

Reviewer #2: Yes: Masanori Atsukawa

Reviewer #3: No

---

## [Editor Report · Acceptance letter]

27 Sep 2019

PONE-D-19-15810R2 

Diagnostic accuracy of midkine on hepatocellular carcinoma: a meta-analysis 

Dear Dr. Yang:

I am pleased to inform you that your manuscript has been deemed suitable for publication in PLOS ONE. Congratulations! Your manuscript is now with our production department. 

With kind regards,

on behalf of

Dr. Wisit Cheungpasitporn 

Academic Editor

PLOS ONE